# Building Dynamic Knowledge Graphs from Text using Machine Reading Comprehension

**Rajarshi Das**[*1], **Tsendsuren Munkhdalai**[2], **Xingdi Yuan**[2], **Adam Trischler**[2], **Andrew McCallum**[1]
[1]College of Information and Computer Sciences
University of Massachusetts, Amherst
{rajarshi, mccallum}@cs.umass.edu
[2]Microsoft Research Montréal
Montréal, Québec, Canada
{tsendsuren.munkhdalai,eric.yuan, adam.trischler}@microsoft.com

## Abstract

We propose a neural machine reading model that constructs dynamic knowledge graphs from procedural text. It builds these graphs recurrently for each step of the described procedure, and uses them to track the evolving states of participant entities. We harness and extend a recently proposed machine reading comprehension (MRC) model to query for entity states, since these states are generally communicated in spans of text and MRC models perform well in extracting entity-centric spans. The explicit, structured, and evolving knowledge graph representations that our model constructs can be used in downstream question answering tasks to improve machine comprehension of text, as we demonstrate empirically. On two comprehension tasks from the recently proposed ProPara dataset (Dalvi et al., 2018), our model achieves state-of-the-art results. The model also outperforms previous approaches on the Recipes dataset (Kiddon et al., 2015), which suggests it may apply broadly to procedural text. Finally, we present some evidence that the model's graphical representations help it to impose commonsense constraints on its predictions.

## 1 Introduction

Automatically building knowledge graphs (KGs) from text is a long-standing goal in artificial intelligence research. KGs organize raw information in a structured form, capturing relationships (labeled edges) between entities (nodes). They enable automated reasoning, e.g., the ability to infer unobserved facts from observed evidence and to make logical "hops," and render data amenable to decades of work in graph analysis.

There exists a profusion of text that describes complex, dynamic worlds in which entities' relationships evolve through time. This includes news articles, scientific manuals, and procedural text (e.g., recipes, how-to guides, and so on). Building KGs from this data would not only help us to study the changing relations among participant entities, but also to make implicit information more explicit. For example, the graphs at each step in Figure 1 help us to infer that the new entity *mixture* is created in the *leaf*, since the previous location of its participant entities (*light, CO₂, water*) was *leaf* – even though this is never stated in the text.

This paper introduces a neural machine-reading model, KG-MRC, that (i) explicitly constructs dynamic knowledge graphs to track state changes in procedural text and (ii) conditions on its own constructed knowledge graphs to improve downstream question answering on the text. Our dynamic graph model is recurrent, that is, the graph at each time step depends on the state of the graph at the previous time step. The constructed graphs are parameterized by real-valued embeddings for each node that change through time.

In text, entities and their states (e.g., their locations) are given by spans of words. Because of the variety of natural language, the same entity/state may be described with several surface forms. To

---

[*]Work performed when author was an intern at MSR Montréal.

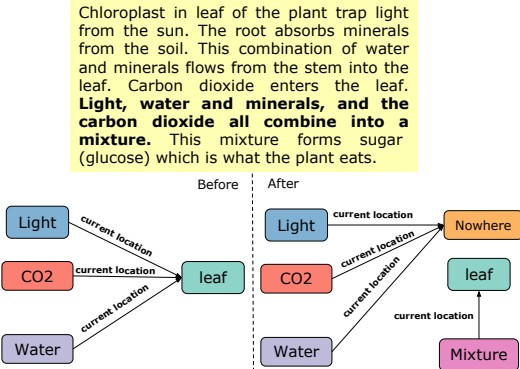

Chloroplast in leaf of the plant trap light from the sun. The root absorbs minerals from the soil. This combination of water and minerals flows from the stem into the leaf. Carbon dioxide enters the leaf. **Light, water and minerals, and the carbon dioxide all combine into a mixture.** This mixture forms sugar (glucose) which is what the plant eats.

Figure 1: Snapshot of the knowledge graphs created by our model before and after reading the sentence in bold-face. Since the KG explicitly stores the current location of *light*, $CO_2$, and *water* as *leaf*, the model can infer that *mixture* is formed in the *leaf* even though this is not explicitly stated. The three participant entities also get destroyed in the process, which is captured in the graph by pointing to a special *Nowhere* node.

address the challenge of entity/state recognition, our model uses a machine reading comprehension (MRC) mechanism (Seo et al., 2017a; Xiong et al., 2017; Chen et al., 2017; Yu et al., 2018, inter alia), which queries for entities and their states at each time step. We leverage MRC mechanisms because they have proven adept at extracting text spans that answer entity-centric questions (Levy et al., 2017). However, such models are static by design, returning the same answer for the same query and context. Since we expect answers about entity states to change over the course of the text, our model's MRC component conditions on the evolving graph at the current time step (this graph captures the instantaneous states of entities).

To address the challenge of aliased text mentions, our model performs *soft co-reference* as it updates the graph. Instead of adding an alias node, like *the leaf* or *leaves* as aliases for *leaf*, the graph update procedure soft-attends (Bahdanau et al., 2014) over all nodes at the previous time step and performs a gated update (Cho et al., 2014; Chung et al., 2014) of the current embeddings with the previous ones. This ensures that state information is preserved and propagated across time steps. Soft co-reference can also handle the case that entity states *do not* change across time steps, by applying a near-null update to the existing state node rather than duplicating it.

At each time step, after the graph has been updated with the (possibly) new states of all entities, our model updates each entity representation with information about its state. The updated information about each individual entity is further propagated to all other entities (§ 4.4). This enables the model to recognize, for example, that entities are present in the same location (e.g., *light, $CO_2$* and *water* in Figure 1). Thus, our model can use the information encoded in its internal knowledge graphs for a more comprehensive understanding of the text. We will demonstrate this experimentally by tackling comprehension tasks from the the recently released PROPARA and RECIPES datasets.

Our complete machine reading model, which both builds and leverages dynamic knowledge graphs, can be trained end-to-end using only the loss from its MRC component; i.e., the negative log-likelihood that the MRC component assigns to the span that correctly describes each entity's queried state. We evaluate our model (KG-MRC) on the above two PROPARA tasks and find that the same model significantly outperforms the previous state of the art. For example, KG-MRC obtains a *9.92%* relative improvement on the hard task of predicting at which time-step an entity moves. Similarly on the latter task, KG-MRC obtains a *5.7%* relative improvement over PROSTRUCT and *41%* relative improvement over other entity-centric models such as ENTNET (Henaff et al., 2017). The same model also obtains state-of-the-art performance on the RECIPES dataset.

## 2  RELATED WORK

There are few datasets that address the challenging problem of tracking entity state changes. The bAbI dataset (Weston et al., 2015) includes questions about movement of entities; however, its language is generated synthetically over a small lexicon, and hence models trained on bAbI often do not generalize well when tested on real-world data. For example, state-of-the-art models like ENTNET (Henaff et al., 2017) and Query Reduction Networks (Seo et al., 2017b) fail to perform well on PROPARA.

PRORead (Berant et al., 2014) introduced the ProcessBank dataset, which contains paragraphs of procedural text as in ProPara. However, this earlier task involves mining arguments and relations from events, not tracking the dynamic state changes of entities. The model that Berant et al. (2014) propose builds small knowledge graphs from the text, but they are not dynamic in nature. The model also relies on densely annotated process structure for training, demanding curation by domain experts. On the other hand, our model, KG-MRC, learns to build dynamic KGs just from annotations of text spans, which are much easier to collect.

For the sentence-level ProPara task they propose, Dalvi et al. (2018) introduce two models: ProLocal and ProGlobal. ProLocal makes local predictions about entities by considering just the current sentence. This is followed by some heuristic/rule-based answer propagation. ProGlobal considers a broader context (previous sentences) and also includes the previous state of entities by considering the probability distribution over paragraph tokens in the previous step. Tandon et al. (2018) recently proposed a neural structured-prediction model, (ProStruct), where hard and soft common-sense constraints are injected to steer their model away from globally incoherent predictions. We evaluate KG-MRC on the two ProPara tasks proposed by Dalvi et al. (2018) and Tandon et al. (2018), respectively, and find that our single model outperforms each of the above models on their respective tasks of focus.

EntNet (Henaff et al., 2017) and query reduction networks (QRN) (Seo et al., 2017b) are two state-of-the-art entity-centric models for the bAbI dataset. EntNet maintains a dynamic memory of hidden states with a gated update to the memory slots at each step. Memory slots can be tied to specific entities, but unlike our model, EntNet does not maintain separate embeddings of individual states (e.g., current locations); it also does not perform explicit co-reference updates. QRN refines the query vector as it processes each subsequent sentence until the query points to the answer, but does not maintain explicit representations of entity states. Neural Process Networks (NPN) (Bosselut et al., 2018) learn to understand procedural text by explicitly parameterizing actions and composing them with entities. These three models return an answer by predicting a vocabulary item in a multi-class classification setup, while in our work we predict spans of text directly from the paragraph.

MRC models have been used previously for extracting the argument of knowledge base (KB) relations, by associating one or more natural language questions with each relation (*querification*). These models have been shown to perform well in a zero-shot setting, i.e., for a previously unseen relation type (Levy et al., 2017), and for extracting entities that belong to non-standard types (Roth et al., 2018). These recent positive results motivate our use of an MRC component in KG-MRC.

## 3 Data & Tasks

We evaluate KG-MRC on the recently released ProPara dataset (Dalvi et al., 2018), which comprises procedural text about scientific processes. The location states of participant entities at each time step (sentence) in these processes are labeled by human annotators, and the names of participant entities are given. As an example, for a process describing photosynthesis, the participant entities provided are: *light, $CO_2$, water, mixture* and *glucose*. Although participant entities are thus known *a priori*, the location of an entity could be *any* arbitrary span in the process text. This makes the task of determining and tracking an entity's changing location quite challenging.

It should also be noted that the dataset does not provide information on whether a particular entity is an input to or output of a process. Not all entities exist from the beginning of the process (e.g. *glucose*) and not all exist at the end (e.g. *water*). Table 1 shows statistics of ProPara. As can be seen, the training set is small, which makes learning challenging.

| # para | 488 |
| # train/#dev/#test | 391/43/54 |
| avg. # entities | 4.17 |
| avg. # sentences | 6.7 |
| # sentences | 3.3K |

Table 1: Statistics of ProPara.

Along with the dataset, Dalvi et al. (2018) introduce the task of tracking state changes at a fine-grained sentence level. To solve this task, a model must answer three categories of questions (10 questions in total) about an entity $E$: (1) Is $E$ created, (destroyed, moved) in the process? (2) When (step #) is $E$ created, (destroyed, moved)? (3) Where is $E$ created, (destroyed, moved from/to)? Cat. 1 asks boolean questions about the existence and movement of entities. Cat. 2 and 3 are harder tasks, as the model must correctly predict the step number at which a state changes as well as the correct locations (text spans) of entities at each step.

Tandon et al. (2018) introduce a second task on the PROPARA dataset that measures state changes at a coarser *process* level. To solve this task, a model must correctly answer the following four types of questions: (1) What are the inputs to the process? (2) What are the outputs of the process? (3) What conversions occur, when and where? (4) What movements occur, when and where? Inputs to a process are defined as entities that exist at the start of the process but not at the end and outputs are entities that exist at the end of the process and were created during it. A conversion is when some entities are created and others destroyed, while movements refer to changes in location. Dalvi et al. (2018) and Tandon et al. (2018) propose different models to solve each of these tasks separately, whereas we evaluate the same model, KG-MRC, on both tasks.

Bosselut et al. (2018) recently released the RECIPES dataset, which has various annotated states (e.g. shape, composition, location, etc.) for ingredients in cooking recipes. We further test KG-MRC on the location task to align with our PROPARA experiments. This is arguably the dataset's hardest task, since it requires classification over more than 260 classes while the others have a much smaller label space (maximum of 4). Note that rather than treating this problem as classification over a fixed lexicon as in previous models, our model aims to find the location-describing span of text in the recipe paragraph.

## 4 MODEL

KG-MRC tracks the temporal state change of entities in procedural text. Naturally, the model is entity-centric (Henaff et al., 2017; Bansal et al., 2017): it associates each participant entity of the procedural text with a unique node and embedding in its internal graph. KG-MRC is also equipped with a neural machine reading comprehension model which is queried about the current location of each entity.

At a high level, the model operates as follows. We summarize some important notation in Table 2. KG-MRC takes as input a paragraph $p = \{w_j\}_{j=1}^P = \{s_t\}_{t=1}^T$, consisting of $P$ tokens spread across $T$ sentences. The model reads this paragraph incrementally. Specifically, at each time step (sentence) $t$, the model reads the paragraph *prefix* comprising all sentences up to and including $s_t$. We then engage the MRC module to query for the state of each participant entity (these participants are known in PROPARA *a priori* and we index them with $i$). The querying process conditions on both the input text and the constructed knowledge graph from the previous time step. In response to a query, the MRC module returns a span from the text that describes the $i$th entity's location at $t$. We encode this into a vector representation. Finally, conditioning on the span vectors for all entities, the model constructs the graph $G_t$ by updating graph $G_{t-1}$ from the previous time step.

The model's knowledge graphs $G_t$ are bipartite, having two sets of nodes with implied connections between them: $G_t = \{e_{i,t}, \lambda_{i,t}\}$. Each node denotes either an entity ($e_{i,t}$) or that entity's corresponding location ($\lambda_{i,t}$), and is associated with a real-valued vector. We use $e_{i,t}$ and $\lambda_{i,t}$ to denote nodes in the graph *and* their vector representations interchangeably. The bipartite graphs $G_t$ have only one (implicit) relation type, the current location, though we plan to extend this in future work. To derive $G_t$ from its previous iterate $G_{t-1}$, we combine both hard and soft graph updates. The update to an entity's node representation with new location information arises from a hard decision made by the MRC model, whereas co-reference between entities across time steps is resolved with soft attention. We now describe all components of the model in detail.

### 4.1 ENTITY AND SPAN REPRESENTATIONS

In the PROPARA dataset, entities appear in the paragraph text.[1] Therefore, we *derive* the initial entity representations from contextualized hidden vectors by encoding the paragraph with a bi-directional LSTM (Hochreiter & Schmidhuber, 1997). This choice has the added advantage that initial entity representations share information through context, unlike in previous models (Henaff et al., 2017; Das et al., 2017; Bansal et al., 2017). Entities in the dataset can be multi-word expressions (e.g., *electric oven*). To obtain a single representation, we concatenate the contextualized hidden vectors corresponding to the start and end span tokens and take a linear projection. i.e., if the mention of entity $i$ occurs between the $j$-th and $j+k$-th position, then the initial entity representation $v_i$ is computed as $v_i = W_e[c_j; c_{j+k}] + b_e$. We use $i$ to index an entity and its corresponding location,

---

[1] We compute the positions of the occurrence of entities by simple string matching.

| Notation | Meaning |
|---|---|
| $N \in \mathbb{N}$ | Number of participant entities in the process. |
| $v_i \in \mathbb{R}^d$ | Initial entity representation, derived from the text, for the $i$-th entity at time $t = 0$ (§ 4.1) |
| $e_{i,t} \in \mathbb{R}^d$ | Entity node representation for the $i$-th entity at time $t$, in the graph $G_t$ (§ 4.4) |
| $\psi_{i,t} \in \mathbb{R}^d$ | Location representation derived from the text for the $i$-th entity at time $t$ (§ 4.1) |
| $\lambda_{i,t} \in \mathbb{R}^d$ | Location node representation for the $i$-th entity at time $t$, in the graph $G_t$ (§ 4.3, 4.4) |
| $\Lambda_t \in \mathbb{R}^{N \times d}$ | Matrix of all location node representations at time $t$. (Essentially all $\lambda_{i,t}$ stacked row-wise at $t$) |
| $U_t \in \mathbb{R}^{N \times N}$ | Soft co-reference matrix at time step $t$ (§ 4.3) |

Table 2: Symbols used in Section 4. The text-based representations of entities and locations are derived from the hidden representations of the context-RNN (§ 4.1). The node representations are added to the graph $G_t$ at the end of time step $t$ (§ 4.4).

while $c_j$ represents the contextualized hidden vectors for token $j$ and $[;]$ represents the concatenate operation. An entity may occur multiple times within a paragraph. We give equal importance to all occurrences by summing the representations for each.

When queried about the current location of an entity, the MRC module (§ 4.2) returns a span of text as the answer, whose representation is later used to update the appropriate node vector in the graph. We obtain this answer-span representation analogously as above, and denote it with $\psi_{i,t}$.

## 4.2 MACHINE READING COMPREHENSION MODEL

Rather than design a specialized MRC architecture, we make simple extensions to a widely used model – DRQA (Chen et al., 2017) – to adapt it to query about the evolving states of entities. In summary, our modified DRQA implementation operates on prefixes of sentences rather than the full paragraph (like PROGLOBAL), and at each sentence (time step) it conditions on both the current sentence representation $s_t$ and the dynamic entity representations in $G_{t-1}$.

For complete details of the DRQA model, we refer readers to the original publication (Chen et al., 2017). Broadly, it uses a multi-layer recurrent neural network (RNN) architecture for encoding both the passage and question text and uses self-attention to match these two encodings. For each token $j$ in the text, it outputs a score indicating its likelihood of being the start or end of the span that answers the question. We reuse all of these operations in our model, modified as described below.

We query the DRQA model about the state of each participant entity at each time step $t$. This involves reading the paragraph up to and including sentence $s_t$. To query, we generate simple natural language questions for an entity, $E$, such as "Where is $E$ located?" This is motivated by the work of Levy et al. (2017). Our DRQA component also conditions on entities. Recall that vector $e_{i,t-1}$ denotes the entity's representation in the knowledge graph $G_{t-1}$. The module conditions on $e_{i,t-1}$ in its output layer, basically the same way as the question representation is used in the *output alignment* step in Chen et al. (2017). However, instead of taking a bi-linear map between the question and passage representations as in that work, we first concatenate the question representation with $e_{i,t-1}$ and pass the concatenation through a 2-layer MLP. This yields an entity-dependent question representation. We use this to compute the output start and end scores for each token position, taking the $\arg\max$ to obtain the most likely span. As mentioned, we encode this span as vector $\psi_{i,t}$ (§ 4.1).

The PROPARA dataset includes two special locations that don't appear as text spans: *nowhere* and *somewhere*. The current location of an entity is *nowhere* when the entity does not exist yet or has been destroyed, whereas it is *somewhere* when the entity exists but its location is unknown from the text. Since these locations don't appear as tokens in the text, the span-predictive MRC module cannot extract them. Following Dalvi et al. (2018), we address this with a separate classifier that predicts, given a graph entity node and the text, whether the entity represented by the node is *nowhere*, *somewhere*, or its location is stated. We learn the location-node representations for *nowhere* and *somewhere* during training.

## 4.3 SOFT CO-REFERENCE

To handle cases when entity states do not change and when states are referred to with different surface forms (either of which could lead to undesired node duplication), our model uses soft co-

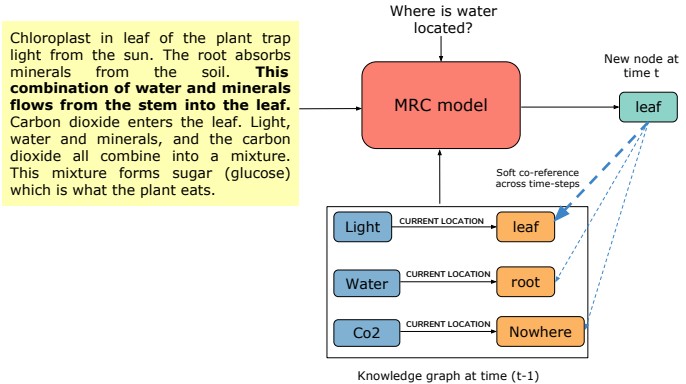

Figure 2: Soft co-reference across time steps. The sentence at the current time step is highlighted. When the MRC model predicts a span (*leaf*) present in the graph at the previous time step, KG-MRC does soft attention and a gated update to preserve information across time steps (§ 4.3). The thicker arrow shows higher attention weight between the old and new node.

reference mechanisms (Figure 2) both across and within time steps. Disambiguation across time steps is accomplished by attention and a gated update, using the incoming location vector $\psi_{i,t}$ and the location node representations from the previous time step:

$$
\begin{aligned}
a_{i,t} &= \text{softmax}(\Lambda_{t-1}\psi_{i,t}) \\
\psi'_{i,t} &= \Lambda_{t-1}^{\top} a_{i,t} \\
g_i &= \text{sigmoid}(W_i[\psi'_{i,t};\psi_{i,t}]+b_i) \\
\lambda'_{i,t} &= g_i\psi_{i,t} + (1-g_i)\psi'_{i,t},
\end{aligned}
\tag{1}
$$

where $\Lambda_{t-1} = [\lambda_{i,t}]_{i=1}^{N} \in \mathbb{R}^{N \times d}$ is a matrix of location node representations from the previous time step (stacked row-wise) and $\psi_{i,t}$ is the location span vector output by the MRC module. The result vector $\lambda'_{i,t}$ is a disambiguated intermediate node representation.

This process only partially addresses node de-duplication. Since different instances of the same location can be predicted for multiple entities, we also perform a co-reference disambiguation within each time step using a self-attention mechanism:

$$
\begin{aligned}
u_{i,t} &= \text{softmax}(\Lambda'_t \lambda'_{i,t}) \\
\lambda_{i,t} &= \Lambda'^{\top}_t u_{i,t},
\end{aligned}
\tag{2}
$$

where $\Lambda'_t = [\lambda'_{i,t}]_{i=1}^{N} \in \mathbb{R}^{N \times d}$ is a matrix of intermediate node representations (stacked row-wise) and $U_t = [u_{i,t}]_{i=1}^{N} \in \mathbb{R}^{N \times N}$ is a co-reference adjacency matrix. We calculate this adjacency matrix at the beginning of each time step to track related nodes within $t$, and re-use it in the graph update step.

## 4.4 GRAPH UPDATE

The graph update proceeds according to the following set of equations for each update layer $l$:

$$
\begin{aligned}
h_{i,t}^{l} &= \text{LSTM}([e_{i,t}^{l-1};\lambda_{i,t}^{l-1};h_{i,t-1}^{l}]) \\
e_{i,t}^{l} &= e_{i,t}^{l-1} + h_{i,t}^{l} \\
\tilde{\lambda}_{i,t}^{l} &= \lambda_{i,t}^{l-1} + h_{i,t}^{l} \\
\lambda_{i,t}^{l} &= \tilde{\Lambda}_t^{l\top} u_{i,t}.
\end{aligned}
\tag{3}
$$

We first compose all connected entity and location nodes with their history summary, $h_{i,t-1}^{l}$, using an LSTM unit. Next, the updated node information is attached to the entity and location representations through two residual updates (He et al., 2016). These propagate information between the entity and location representations; i.e., if two entities are at the same location, then the corresponding entity representations will receive a similar update. Likewise, location representations are updated with pertinent entity information. Last, we perform another co-reference pooling operation for the

location nodes. After the recurrent and residual graph updates, information propagation may yield different, diverging representations for nodes that belong to the same location. This final pooling operation corrects for this, by tying the co-referent representations in a soft way. It uses the previously computed adjacency matrix $U_t$ and $\tilde{\Lambda}_t^l$, which is a row-wise stacked matrix of the $\tilde{\lambda}_{i,t}^l$.

The recurrent graph module stacks $L$ such layers to propagate node information along the graph's edges. The resulting node representations are $e_{i,t}^L$ and $\lambda_{i,t}^L$ for each participant entity and its location. We use $e_{i,t} = e_{i,t}^L$ to condition the MRC model, as described in §4.2. We make use of this particular graph module structure, rather than adopting an existing model like GraphCNNs (Edwards & Xie, 2016; Kipf & Welling, 2017), because recurrent networks are designed to propagate information through time.

## 4.5 Training

The full KG-MRC model is trained end-to-end by minimizing the negative log-likelihood of the correct span tokens under the MRC module's output distribution and the textual entailment model. This is a fairly soft supervision signal, since we do not train the graph construction modules directly. We teacher-force the model at training time by updating the location-node representations with the encoding of the correct span. We do not pretrain the MRC module, but we represent paragraph tokens with pretrained FastText embeddings (Joulin et al., 2016). See the appendix A for full implementation and training details.

## 5 Experiments and Discussion

We evaluate our model on three different tasks. We also provide an ablation study along with quantitative and qualitative analyses to highlight the performance contributions of each module.

### 5.1 Results on Procedural Text

We benchmarked our model on two PROPARA comprehension tasks introduced respectively in Dalvi et al. (2018) and Tandon et al. (2018). Refer to Section 3 for a detailed description about the data and tasks. Dalvi et al. (2018) and Tandon et al. (2018) respectively introduce a specific model for each task, whereas we test KG-MRC on both tasks. A primary motivation for building KGs is because they can be *queried* for salient knowledge in downstream applications. We evaluate KG-MRC on the above two tasks by querying the KGs it builds at each time-step; we use the official evaluation pipeline[2] for each task. In results below, we report an average score of three runs of our model with different hyperparameter settings.

#### 5.1.1 Task 1: Sentence-level Evaluation

Table 3 shows our main results on the first task. Following the original task evaluation, we report model accuracy on each subtask category and macro and micro averages over the subtasks.

Human performance is 79.69%, micro-average. A state-of-the-art memory augmented network, ENTNET (Henaff et al., 2017), which is built to track entities but lacks an explicit graph structure, achieves 25.96%. The previous best performing model is PROGLOBAL, which achieves 45.37%. Our KG-MRC improves over this result by 1.25% absolute score in terms of micro-averaged accuracy. Comparing various models for each subtask category, PROGLOBAL leads in Category 1 by a small margin of around 0.1%. For the more challenging Categories 2 and 3, KG-MRC outperforms PROGLOBAL by a large margin. These questions require fine-grained predictions of state changes.

---

[2]https://github.com/allenai/propara/tree/master/propara/eval

|  | Cat 1 | Cat 2 | Cat 3 | Macro-avg | Micro-avg |
|---|---|---|---|---|---|
| Human upper bound | 91.67 | 87.66 | 62.96 | 80.76 | 79.69 |
| Majority | 51.01 | – | – | – | – |
| Rule based | 57.14 | 20.33 | 2.40 | 26.62 | 26.24 |
| Feature based | 58.64 | 20.82 | 9.66 | 29.7 | 29.64 |
| EntNet (Henaff et al. (2017)) | 51.62 | 18.83 | 7.77 | 26.07 | 25.96 |
| Pro-Local (Dalvi et al. (2018)) | 62.65 | 30.50 | 10.35 | 34.50 | 33.96 |
| Pro-Global (Dalvi et al. (2018)) | **62.95** | 36.39 | 35.90 | 45.08 | 45.37 |
| KG-MRC (ours) | 62.86 | **40.00** | **38.23** | **47.03** | **46.62** |

Table 3: Task 1 results (accuracy).

### 5.1.2 TASK 2: DOCUMENT-LEVEL EVALUATION

We report the performance of our model on the document-level task, along with previously published results, in Table 4. The same KG-MRC model achieves 3.02% absolute improvement in $F_1$ over the previous best result of PROSTRUCT. PROSTRUCT incorporates a set of commonsense constraints for globally consistent predictions. We analyzed KG-MRC's outputs and were surprised to discover that our model learns these commonsense constraints from the data in an end-to-end fashion, as we show quantitatively in §5.4.

|  | Precision | Recall | $F_1$ |
|---|---|---|---|
| Pro-Local (Dalvi et al. (2018)) | **77.4** | 22.9 | 35.3 |
| QRN (Seo et al. (2017b)) | 55.5 | 31.3 | 40.0 |
| EntNet (Henaff et al. (2017)) | 50.2 | 33.5 | 40.2 |
| Pro-Global (Dalvi et al. (2018)) | 46.7 | **52.4** | 49.4 |
| Pro-Struct (Tandon et al. (2018)) | 74.2 | 42.1 | 53.75 |
| KG-MRC (ours) | 64.52 | 50.68 | **56.77** |

Table 4: Task 2 results.

### 5.2 RECIPE DESCRIPTION EXPERIMENTS

We also evaluate our model on the RECIPES dataset, for which we predict the evolving locations of cooking ingredients. In the original work of Bosselut et al. (2018), they treat this problem as classification over a fixed lexicon of locations. KG-MRC searches for the correct location span in the text. On this task, our model outperforms the baseline NPN model by a significant margin, achieving a score of **54.27**% $F_1$ compared to NPN's 51.28% $F_1$. On further analysis of the results, we found several cases where our model was wrongly penalized, e.g., for predicting the span "saucepan" when the ground truth class label was "pan." We believe that our results would improve further if we mapped our predicted spans to the ground truth class labels.

### 5.3 ABLATION STUDY

We performed an ablation study to evaluate different model variations on PROPARA Task 1. The main results are reported in Table 5. Removing the soft co-reference disambiguation within time steps (Equations 2) from KG-MRC resulted in around 1% performance drop. The drop is more significant when the co-reference disambiguation across time steps (Equations 1) is removed.

We also replaced the recurrent graph module with the standard LSTM unit and used the LSTM hidden state for the entity representation. Because this model variant does not propagate information across graph nodes (the final step in Equations 3), we observed a large performance decrease.

For the last two variations, we simply train the MRC model in isolation and predict location spans from the current sentence or paragraph prefix text (i.e., the current and all previous sentences). These models construct no internal knowledge graphs. We can see that training the MRC model on paragraph prefixes already provides a good starting performance of 40.83% micro-average, which is significantly boosted by the recurrent graph module and graph conditioning up to 47.64%.

| | Cat 1 | Cat 2 | Cat 3 | Macro-avg | Micro-avg |
|---|---|---|---|---|---|
| KG-MRC | 58.55 | 38.52 | 42.22 | 46.43 | 47.64 |
| - Coref across time steps | 61.07 | 37.38 | 35.58 | 44.68 | 46.32 |
| - Coref within time step | 57.88 | 38.09 | 40.19 | 45.39 | 46.63 |
| - Coref in the graph-update step | 60.91 | 34.71 | 32.34 | 42.65 | 44.48 |
| Standard LSTM as graph unit | 56.84 | 13.15 | 10.95 | 26.98 | 29.97 |
| MRC on entire paragraph | 58.85 | 21.82 | 26.52 | 35.73 | 35.98 |
| MRC on prefix | 61.28 | 32.58 | 29.48 | 41.11 | 40.83 |

Table 5: Ablation experiment results

## 5.4 COMMONSENSE CONSTRAINTS

For accurate, globally consistent predictions on the second PROPARA task, Tandon et al. (2018) introduced a set of commonsense constraints that they impose on their model in a pruning stage. Stated in natural language, these constraints are: 1) An entity must **exist** before it can be **moved** or **destroyed**; 2) An entity cannot be **created** if it already **exists**; 3) An entity cannot **change** until it is **mentioned** in the paragraph.

To analyze whether our model can learn these constraints directly from data, we count the number of model predictions that violate constraints on the test set. To our surprise, this demonstrates that KG-MRC learns to violate fewer constraints (proportionally) than PROSTRUCT, even without explicitly training it to do so. In more detail, we find that KG-MRC, like PROSTRUCT, does not violate any Type 1 or Type 2 constraints. In Table 6 we compare several models in terms of Type 3 constraint violations. Note that we only count instances where a model predicts an entity state change.

| Model | State Change Predictions | Violations | Violation Proportion (%) |
|---|---|---|---|
| PROSTRUCT (Tandon et al. (2018)) | 270 | 17 | 6.30 |
| MRC on entire paragraph | 381 | 104 | 27.30 |
| MRC on prefix | 703 | 154 | 21.93 |
| Standard LSTM as graph unit | 447 | 20 | 4.47 |
| KG-MRC | 466 | 19 | **4.08** |

Table 6: Commonsense constraint violations.

As shown, KG-MRC makes fewer Type 3 violations that PROSTRUCT. Furthermore, MRC models without recurrent graph modules perform worse in terms of constraint violations than both KG-MRC and a model using a standard LSTM as its graph unit. This suggests that recurrent graphical representations play an important role in helping the model to learn and adhere to the constraints.

## 5.5 QUALITATIVE ANALYSIS

We picked an example from the test data and took a closer look at the model outputs to investigate how KG-MRC dynamically adjusts its decisions via the dynamic graph module and finds accurate spans with the conditional MRC model. The step-by-step output of both PROGLOBAL (Dalvi et al. (2018)) and KG-MRC is shown in Table 7, where we track the state of entity *blood* across six sentences. KG-MRC outputs smoother and more accurate predictions.

| Sentences | Location of entities after each sentence | |
|---|---|---|
| (Before first sentence) | somewhere | somewhere |
| Blood enters the right side of your heart. | heart | right side of your heart |
| Blood travels to the lungs. | lung | lungs |
| Carbon dioxide is removed from the blood. | blood | lungs |
| Oxygen is added to your blood. | lung | lungs |
| Blood returns to left side of your heart. | blood | heart |
| The blood travels through the body. | body | body |

Table 7: Two models' predictions of entity locations, on randomly selected paragraph about blood circulation. In this example the entity is **blood**. Predicted results from Pro-Local (Dalvi et al. (2018)) are in orange, results from KG-MRC are in red, important locations in paragraph are in blue.

## 6 CONCLUSION

We proposed a neural machine-reading model that constructs dynamic knowledge graphs from text to track locations of participant entities in procedural text. It further uses these graphical representations to improve its downstream comprehension of text. Our model, KG-MRC, achieves state-of-the-art results on two question-answering tasks from the PROPARA dataset and one from the RECIPES dataset. In future work, we will extend the model to construct more general knowledge graphs with multiple relation types.

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

# A IMPLEMENTATION DETAILS

Implementation details of KG-MRC are as follows.

In all experiments, the word embeddings are initialized with FastText embeddings (Joulin et al., 2016); we use a document LSTM with two layers, the number of hidden units in each layer is 64. We apply dropout rate of 0.4 in all recurrent layers, and 0.3 in all other layers. The number of recurrent graph layers were set to ($L = 2$). The hidden unit size for the recurrent graph component was set to 64.

During training, the mini-batch size is 8. We use *adam* (Kingma & Ba, 2014) as the step rule for optimization, The learning rate is set to 0.002. The model is implemented using *PyTorch* (Paszke et al., 2017).

