# OpenReview forum: "Building Dynamic Knowledge Graphs from Text using Machine Reading Comprehension"
_ICLR.cc/2019/Conference_

### Official Review · AnonReviewer2 · 2018-11-03
**Contributions are novel and convinced about significance**

**Rating:** 7
**Confidence:** 4

**Review:**

The paper addresses a challenging problem of predicting the states of entities over the description of a process. The paper is very well written, and easily understandable. The authors propose a graph structure for entity states, which is updated at each step using the outputs of a machine comprehension system. The approach is novel and well motivated. I will suggest a few improvements:

1. the NPN model seems a good alternative, will be good to have a discussion about why your model is better than NPN. Also, NPN can probably be modified to output spans of a sentence. I will be curious to know how it performs.

2. A more detailed illustration of the system / network is needed. Would have made it much easier to understand the paper.

3. What are the results when using the whole training set of Recipes ?

---

> ### Author Response · Authors · 2018-11-21
> **Response to Reviewer 2 comments**
>
> We’re glad that you found the paper interesting and well-written. To address your comments and questions:
>
> 1. the NPN model seems a good alternative, will be good to have a discussion about why your model is better than NPN. Also, NPN can probably be modified to output spans of a sentence. I will be curious to know how it performs.
>
> The NPN model requires a pre-defined lexicon of action types (i.e., verbs), such as cut, bake, boil, etc. For the recipes dataset, the action types and their causal effects were manually collected and defined. Since the ProPara dataset does not have these annotations, we would have to manually identify action types to apply NPN to it.
> Also, NPN treats the state change as a classification problem (of about 260 classes that are also manually defined). In contrast, KG-MRC finds the state-describing span in the text directly, which we believe is a more generic approach.
>
> 2. A more detailed illustration of the system / network is needed. Would have made it much easier to understand the paper.
>
> We agree that more detail would help readers to understand the model better. We’ve made some hopefully significant updates to Section 4 (model description and notation) to improve clarity, and we hope you’ll take the time to read the new manuscript. Two important additions are a high-level summary of the model, which we give at the beginning of Section 4, and a table (Table 2) that lists what each symbol represents along with its dimensions.
>
> 3. What are the results when using the whole training set of Recipes ?
>
> We’ve completed an experiment on the full Recipes dataset and updated the paper to describe the result (this experiment did not finish in time for the initial submission). The model’s F1 score improves from 51.64 on the partial data to 54.27 on the full data, surpassing the previous state of the art by a more significant margin.

---

### Official Review · AnonReviewer1 · 2018-11-05
**Meaningful contribution, but hard to read**

**Rating:** 7
**Confidence:** 4

**Review:**

The paper proposes a recurrent knowledge graph (bipartite graph between entities and location nodes) construction & updating mechanism for entity state tracking datasets such as (two) ProPara tasks and Recipes. The model goes through the following three steps: 1) it reads a sentence at each time step t and identifies the location of each entity via machine reading comprehension model such as DrQA (entities are predefined). 2) Co-reference module adjusts relationship scores (soft adjacency matrix) among nodes, including possibly new nodes introduced by the MRC model. 3) to propagate the relational information across all the nodes, the model performs L layers of LSTM for each entity that attend on other nodes via attention (where the weights come from the adjacency matrix). The model repeats the three steps for each sentence. The model is trained by directly supervising for the correct span by the MRC model at each time step, which is possible because the data provides strong supervision for each sentence (not just the answer at the end).
 The model achieves the state of the art in the two tasks of ProPara and Recipes dataset.

Strengths: The paper provides an elegant solution for tracking relationship between entities as time (sentence) progresses. I also agree with the authors that this line of work (dynamic KG construction and modification) is an important area of research. While the model shares a similar spirit to EntNet, I think the model has enough distinctions / contributions, especially given that it outperforms EntNet by a large margin. The model also obtains non-trivial improvement over previous SOTA models.

Weaknesses: Paper could have been written better. I had hard time understanding it. The notations are overall confusing and not explained well. Also there are a few unclear parts which I discuss in questions below.

Questions:
1. Are e_{i,t} and lambda_{i,t} vectors? Scalars? Abstract node notations? It is not clear in the model section. Also, it took me a long time to figure out that ‘i’ is used to index each entity (it is mentioned later).
2. The paper says v_i (initial representation of each entity) is obtained by looking at the contextualized representations (LSTM outputs) of entity mention in the context. What happens if there are multiple mentions in the text? Which one does it look at?
3. For the LSTM in the graph update, why does it have only one input? Shouldn’t it have two inputs, one for previous hidden state and the other for input?
4. Regarding Recipe experiments, the paper says it reaches a better performance than the baseline using just 10k examples out of 60k. This is great, but could you also report the number when the full dataset is used?
5. What does it mean that in training time the model “updates” the location node representation with the encoding of correct span. Do you mean you use the encoding instead?
6. For ProPara task 2, what threshold did you choose to obtain the P/R/F1 score? Is it the threshold that maximizes F1?

---

> ### Author Response · Authors · 2018-11-21
> **Response to Reviewer 1 comments**
>
> Thank you for the useful feedback. We’ve updated our paper to take it into account -- we’ve updated the model description and the notation in Section 4 to clarify our method. Two important additions are a high-level summary of the model, which we give at the beginning of Section 4, and a table (Table 2) that lists what each symbol represents along with its dimensions. We also made several updates that address your specific questions.
>
> 1. Are e_{i,t} and lambda_{i,t} vectors? Scalars? Abstract node notations? It is not clear in the model section. Also, it took me a long time to figure out that ‘i’ is used to index each entity (it is mentioned later).
>
> The entity and location embeddings  e_{i,t} and lambda_{i,t} are d-dimensional vectors, although we also overload the symbols to refer to abstract nodes in the model’s knowledge graphs. In the updated manuscript we state both these facts explicitly and state much earlier that ‘i’ is the index for entities.
>
> 2. The paper says v_i (initial representation of each entity) is obtained by looking at the contextualized representations (LSTM outputs) of entity mention in the context. What happens if there are multiple mentions in the text? Which one does it look at?
>
> When there are multiple mentions of entity i, the initial representation v_i is formed by summing the representations of each mention. We have updated the paper to clarify this (Sec 4.1).
>
> 3. For the LSTM in the graph update, why does it have only one input? Shouldn’t it have two inputs, one for previous hidden state and the other for input?
>
> Good point! We’ve improved the notation used to describe the model in Section 4. The update equation now shows clearly that the LSTM takes in the concatenation of two node inputs (entity and location embeddings) along with the previous hidden state.
>
> 4. Regarding Recipe experiments, the paper says it reaches a better performance than the baseline using just 10k examples out of 60k. This is great, but could you also report the number when the full dataset is used?
>
> We’ve completed an experiment on the full Recipes dataset and updated the paper to describe the result (this experiment did not finish in time for the initial submission). The model’s F1 score improves from 51.64 on the partial data to 54.27 on the full data, surpassing the previous state of the art by a more significant margin.
>
> 5. What does it mean that in training time the model “updates” the location node representation with the encoding of the correct span. Do you mean you use the encoding instead?
>
> We meant that we perform teacher-forcing to train the model. During training, we extract the context encodings for the groundtruth span and use these in downstream operations  to obtain the node representations. At test time, we use the MRC module’s predicted span rather than the groundtruth.
>
> 6. For ProPara task 2, what threshold did you choose to obtain the P/R/F1 score? Is it the threshold that maximizes F1?
>
> For ProPara task 2, our model was optimized for micro averaged F1 on the development set. Tandon et al. (2018) were kind enough to provide us with their evaluation script.

---

> > ### Comment · AnonReviewer1 · 2018-12-02
> > **Thanks for clarification**
> >
> > Thank you for the clarification. Now the paper seems to be more clear than before. As my major concern was readability and it is now enhanced, I am raising my score to 7. Great work.

---

### Official Review · AnonReviewer3 · 2018-11-05
**Good ideas and results, could use some work with explanation**

**Rating:** 6
**Confidence:** 4

**Review:**

* Summary
This paper addresses machine reading tasks involving tracking the states of entities over text. To this end, it proposes constructing a knowledge graph using recurrent updates over the sentences of the text, and using the graph representation to condition a reading comprehension module. The paper reports positive evaluations on three different tasks.

* Review

This is an interesting paper. The key technical component in the proposed approach is the idea that keeping track of entity states requires (soft) coreference between newly read entities and locations and the ones existing in the knowledge graph constructed so far.

The proposed method seems plausible, but some details are impressionistic and it is not clear why and whether the modeling choices do what the paper says. This is especially the case in a few places involving coreference:
1. The paper says at the top of page 6 that the result of Eq 1 is a disambiguated intermediate node representation.
2. The self attention in Eq 2 performs coreference disamguation which prevents different instances of the same location from being predicted for multiple entities.

While these may indeed be working as advertised, it would be good to see some evaluation that verifies that after learning, what is actually happening is coreference.

Why does the graph update require coreference pooling again?  Don't the updates in Eq 1 and 2 take care of this? The ablation does not test this, right?

Another modeling choice that is not clear is regarding how the model processes the text -- reading prefixes of the paragraph, rather than one sentence at a time. What happens if the model is changed to be read one sentence at a time?

That the model implicitly learns constraints from data is interesting!

Bottomline: The paper presents interesting ideas and good results, but would be better if the modeling choices were better explored/motivated.

---

> ### Author Response · Authors · 2018-11-21
> **Response to Reviewer 3 comments**
>
> Thanks for the insightful comments. We’ve tried to improve our paper based on your feedback. Most significantly, we’ve performed additional ablation studies to confirm that our modeling choices improve performance, and we provide further empirical insight on what the coreference operations do. We’ve also updated the model description and the notation in Section 4 to clarify modeling mechanisms and choices. Two important additions are a high-level summary of the model, which we give at the beginning of Section 4, and a table (Table 2) that lists what each symbol represents along with its dimensions. Below we address your concerns point-by-point.
>
> The proposed method seems plausible, but some details are impressionistic and it is not clear why and whether the modeling choices do what the paper says. This is especially the case in a few places involving coreference:
> 1. The paper says at the top of page 6 that the result of Eq 1 is a disambiguated intermediate node representation.
> 2. The self attention in Eq 2 performs coreference disamguation which prevents different instances of the same location from being predicted for multiple entities.
> While these may indeed be working as advertised, it would be good to see some evaluation that verifies that after learning, what is actually happening is coreference.
> ======
> Based on your comments, we’ve performed additional ablations to measure the impact of the co-reference mechanisms. We find that removing any of them leads to a decrease in performance (Rows 2, 3, 4 of Table 5).
>
> To provide more than just this quantitative insight, we’ll expand here on how KG-MRC handles coreference to better motivate the modeling choices:
> The construction of graph G_t from G_{t-1} uses co-reference disambiguation of nodes to prevent node duplication and to enforce temporal dependencies. We perform coreference disambiguation between location nodes of G_t and G_{t-1} via Eq. 1 (call this inter-graph coreference) and between the location nodes in the same graph Gt (call this intra-graph coreference) via Eq. 2. The inter-graph coreference yields new, intermediate representations for the nodes in G_t. These are further updated via the intra-graph coreference step.
>
> Inter-graph Co-ref: One way to think about this is that we construct a new graph G_t at every time step. Now the graph G_{t-1} might contain some location nodes which are predicted again at time step ‘t’ (e.g., in Figure 2, leaf node already existed in G_{t-1}). Instead of replacing an old node with an entirely new node at ‘t’, we take a recurrent approach and do a gated update that preserves some information stored in the node in previous time steps while adding new information unique to time step ‘t’.
>
> Intra-graph Co-ref: Inter-graph co-ref isn’t enough since the MRC module makes its span predictions independently. This means that, at time step t, the model could predict the same span/location for multiple entities and add all these duplicates to the graph. Moreover, a single location might have the same surface form but be from different parts of the paragraph (e.g. “leaf” in the 1st and the 5th sentence of the para in figure 2). The operations in Eq. 2 resolve this by performing self-attention (i.e., the predicted locations of all entities are compared to each other).
> =====

---

> > ### Author Response · Authors · 2018-11-21
> > **Response to Reviewer 3 comments (contd.)**
> >
> > Response continued from above.
> >
> > Why does the graph update require coreference pooling again?  Don't the updates in Eq 1 and 2 take care of this? The ablation does not test this, right?
> > =====
> > We agree that the coreference pooling in the graph update seems repetitive at first glance. We have further clarified the explanation given in the text and included another ablation experiment  (row 4 of Table 5) to confirm its usefulness.  This step does indeed repeat Eq. 2. In a nutshell, this is necessary because, after the recurrent and residual graph updates (Eqs 3.1 - 3.3) that propagate information across edges, we may end up with different representations for location nodes corresponding to the same location. We don’t want these representations to diverge from each other because of information propagation.
> >
> > To give you more detail:
> > The graph update step ensures information propagation between entities and location representations. Specifically if the current location of entity “e_t” is predicted as “\lambda_t”, the graph update steps ensures that both the entity and location representation gets the same update (via eq 3.2 and 3.3). This would have been sufficient if every entity had a unique location. But, multiple entities can actually exist in the same location. Let’s consider this small graph below
> >
> > Water - -> leaf
> > CO_2 --> leaf
> >
> > Here both water and CO_2 exist in the same location, leaf.  But let’s say that the MRC model picked the “leaf” span from sentence 1 (of the text in Fig 2) for “Water” and from sentence 4 for CO_2. In reality, they refer to the same location entity “leaf”. Now, due to eq. 3.3, the two embeddings of leaf will get two different residual updates (one would be corresponding to Water and other would be because of CO_2). Because of the different updates, the two representations of the same entity might diverge. To remedy this, we re-use the coreference matrix “U” we create in eq. (2), which should already have a high attention score corresponding to the two leaf locations. Thus we perform a similar operation to the intra-graph update.
> > ====
> > Another modeling choice that is not clear is regarding how the model processes the text -- reading prefixes of the paragraph, rather than one sentence at a time. What happens if the model is changed to be read one sentence at a time?
> > ====
> > The “prefixes” that our model reads at each time step comprise all sentences up to and including the current sentence s_t. The motivation for this modeling choice was empirical. In our preliminary experiments we evaluated alternative strategies, such as (a) only considering the current sentence s_t, and (b) considering the entire paragraph at every time step. We found that operating on prefixes performed best. This is in line with the findings of Dalvi et al., 2018, where the Pro-Global model (which uses prefixes) performs better than the Pro-Local model (which operates on single sentences).

---

### Author Response · Authors · 2018-11-20
**Summary of updates**

Based on the insightful feedback from our reviewers, we’ve updated our paper and believe it is substantially improved. Below we summarize the general changes, and in responses to individual reviewers, we respond directly to their comments/questions.

Full Recipes experiment (Section 5.2):
We completed an experiment on the full Recipes dataset and updated the paper with the result. This experiment did not finish in time for the initial submission, so we only had results from a model trained on partial data. The model’s F1 score improved from 51.64 on the partial data to 54.27 on the full data, surpassing the previous state of the art (51.27) by a more significant margin.

Additional ablations (Table 5):
To demonstrate more clearly the impact of several modelling choices, we’ve completed additional ablation experiments. Specifically, these measure the performance contributions of the model’s coreference operations and show that they are important.

Update to results on commonsense constraints (Table 6):
After submission, we discovered a string-matching bug in the script that calculates commonsense constraint violations. Correcting this bug changes our results slightly, although the general takeaway is the same. KG-MRC still does not violate any commonsense constraints of Types 1 and 2 (as defined in ProStruct (Tandon et al., 2018)), but we find that both our model and ProStruct violate a small number of Type 3 constraints -- KG-MRC notably makes proportionally fewer violations than ProStruct (4.1% vs 6.3%). We also report violation numbers for several ablated variants of our model and find that they consistently perform worse than the full model. These results are all summarized in Table 6 of the updated manuscript.

Improved Section 4
We received feedback that additional details and notational changes in the model description would help readers to understand the model better. We, therefore, made some hopefully significant updates to Section 4 to improve clarity. Two important additions are a high-level summary of the model, which we give at the beginning of Section 4, and a table (Table 2) that lists what each variable represents along with its dimensions.

---

### Meta-Review · Area_Chair1 · 2018-12-14
**An interesting task & models with solid empirical results.**

**Confidence:** 4
**Recommendation:** Accept (Poster)

**Metareview:**

This paper investigates a new approach to machine reading for procedural text, where the task of reading comprehension is formulated as dynamic construction of a procedural knowledge graph. The proposed model constructs a recurrent knowledge graph (as a bipartite graph between entities and location nodes) and tracks the entity states for two domains: scientific processes and recipes.

Pros:
The idea of formulating reading comprehension as dynamic construction of a knowledge graph is novel and interesting. The proposed model is tested on two different domains: scientific processes (ProPara) and cooking recipes.

Cons:
The initial submission didn't have the experimental results on the full recipe dataset and also had several clarity issues, all of which have been resolved through the rebuttal.

Verdict:
Accept. An interesting task & models with solid empirical results.